# ^11^C- and ^18^F-Radiotracers for In Vivo Imaging of the Dopamine System: Past, Present and Future

**DOI:** 10.3390/biomedicines9020108

**Published:** 2021-01-22

**Authors:** Michael R. Kilbourn

**Affiliations:** Department of Radiology, University of Michigan Medical School, Ann Arbor, MI 48105, USA; mkilbour@umich.edu

**Keywords:** positron emission tomography, radiotracer, dopamine, receptor, transporter

## Abstract

The applications of positron emission tomography (PET) imaging to study brain biochemistry, and in particular the aspects of dopamine neurotransmission, have grown significantly over the 40 years since the first successful in vivo imaging studies in humans. In vivo PET imaging of dopaminergic functions of the central nervous system (CNS) including dopamine synthesis, vesicular storage, synaptic release and receptor binding, and reuptake processes, are now routinely used for studies in neurology, psychiatry, drug abuse and addiction, and drug development. Underlying these advances in PET imaging has been the development of the unique radiotracers labeled with positron-emitting radionuclides such as carbon-11 and fluorine-18. This review focuses on a selection of the more accepted and utilized PET radiotracers currently available, with a look at their past, present and future.

## 1. Introduction

The development of positron emission tomography (PET) during the past almost 60 years has provided unique and valuable methods for probing the mysteries of the human brain, that 1.4 kg (about three-pound) organ that resists for the most part our desires to dissect, inspect, and understand the unknown. Success in PET imaging is truly multidisciplinary, requiring combinations of physicians and scientists from many fields (chemistry, physics, mathematics, pharmacology, physiology, engineering, medicine, and others). Forming the foundation of PET imaging, however, are the molecules—the chemicals that, when appropriately selected or sometimes even designed, are prepared in radiolabeled form and utilized to create the in vivo images that hopefully represent the biochemistry of the living human brain. An impressively large number of molecular species have now been synthesized with radionuclides suitable for PET imaging, but in truth most have failed as in vivo radiotracers; a small number have shown encouraging results in animal studies but were never advanced to human studies; an even smaller subset was successfully translated to human imaging studies; finally, a very small selection have reached the pinnacle of being routinely used by multiple research laboratories around the world, and most recently for patient care in clinical settings.

## 2. In Vivo PET Radiotracers: General Considerations

What are the characteristics of a successful in vivo PET radiotracer for imaging neurochemistry in the human brain? The first and most obvious is sufficient brain uptake, which can be through either facilitated movement by a membrane transporter, or passive permeation of the blood–brain barrier (BBB) by lipophilic molecules. Most radiotracers are small molecules (MW < 500 Da) that are designed to be passively permeable, with a “sweet spot” of lipophilicity (in the approx. log P range of 1.5–3.5) to maximize brain uptake but not produce excessive nonspecific background distributions. Compounds chosen should have good selectivity (or even specificity) for the biochemical target of interest, although as will be discussed later, at times nonselective radiotracers are of use where nature has fortuitously segregated the possible targets. In practice, it has been determined that most if not all successful radioligands intended to bind to enzymes, receptors, or transporters demonstrate low nanomolar values for in vitro binding affinities. The best radiotracers will have appropriate in vivo pharmacokinetics in the brain, such that analytical methods employing pharmacokinetic models can be applied to extract numerical estimates of the targeted biochemical process. Of course, candidate compounds must be radiolabeled: fortunately, use of carbon-11 (T_1/2_ = 20.4 min) allow the potential synthesis of any organic molecule, and as fluorine is commonly used in medicinal chemistry [1], fluorine-18 (T_1/2_ = 110 min) is a highly useful substituent for PET radiotracers. The development of new and improved methods for radiolabeling with these two radionuclides over the last five decades has resulted in an ability to radiolabel a very large selection of drugs and related compounds in the required yields, purities, and molar activities (radioactivity/mass) for human applications. Finally, the radiotracers should not be actively extruded from the brain by efflux transporters (e.g., p-glycoprotein) or form radiolabeled metabolites that can interfere with the interpretation of the PET images [2]. Examples of several important dopaminergic PET radiotracers are shown in Table 1 and discussed in the following sections. It is important to note that of these commonly used PET radiopharmaceuticals, only [^11^C]methylphenidate (Ritalin^®^) and [^11^C]deprenyl (Selegiline^®^) are simply radiolabeled forms of a clinical drug.

## 3. Dopamine and PET Imaging in the Brain: The Big Picture

The dopaminergic system of the human brain was a very early and prominent target for in vivo radiotracer development. It has been demonstrated or implicated to be important in a wide variety of neurologic and psychiatric diseases, but it is abundantly clear that no single common change in the actions of dopamine explains all pathologies. Some are undoubtedly a result of combinations of changes in multiple aspects of the life span of dopamine: its birth (biosynthesis), travels (transport, synaptic release), activities (receptor binding), and death (metabolism). As no single step in dopamine neurotransmission would serve all in vivo imaging needs, what has evolved through the years is a holistic approach to probing the system, and as seen in Table 1 nearly all aspects of the dopaminergic system have been targets of PET imaging. It is worth noting that, with the exception of the radiotracers that have been developed for study of dopamine biosynthesis, the radioligands in Table 1 show high specificity for the type of target: receptor ligands have poor affinities for the transporters, transporter ligands do not bind to the receptors, and the ligands for the transporters are specific for either the DAT (neuronal membrane dopamine transporter) or the VMAT2 (vesicular monoamine transporter type 2). This specificity is crucial to interpreting in vivo PET images that, in reality, look very similar (Figure 1) for many of the radioligands.

As radiotracers were developed, many questions have arisen regarding their purpose for in vivo imaging of the dopamine system. What should we study, and are different properties more or less important in different diseases? For example, should we only be interested in radiotracers that inform on the concentration of dopamine in brain tissue, or numbers of the many important proteins (enzymes, transporters, and receptors) involved in dopaminergic neurotransmission? Or should we concentrate on what might be functional radiotracers, which inform us on how well these various proteins are operating in pathological versus normal brain tissues? Or is it more beneficial if we develop radiotracers useful in evaluating how drugs, old and new, interact with the dopaminergic system so that we might improve our therapies? Will one radiotracer suffice for these different applications, or will it require multiple radiotracers tailored to use? Finally, it has also become evident that the properties needed for successful in vivo radiotracers may not be uniform between different targeted binding sites: concentrations of any single binding site may vary considerably between brain regions (e.g., dopamine receptors in caudate-putamen vs. thalamus and cortex), may be different for subtypes (D2 vs. D3 in thalamus), and even larger differences are observed between classes of binding sites (>10-fold higher VMAT2 than dopamine receptors in striatum) [3]. These many questions have been in part responsible for the development of new radiotracers that continues today. In some areas, optimum radiotracers remain to be discovered.

From a historical perspective, not everything was tried at once; development efforts for PET radiopharmaceuticals useful for all of the varied aspects of the brain dopamine system have lasted almost 50 years, and a comprehensive listing of all dopaminergic PET radiotracers is beyond the scope of this review. Research into new radiopharmaceuticals continues, with new radiolabeled compounds being actively investigated in preclinical animal models; the interested reader is encouraged to access the primary scientific literature to learn more. Undoubtedly, the continued efforts in radiopharmaceutical development will fuel the push of PET imaging into basic research, pharmaceutical development, and patient care.

## 4. Dopamine Biosynthesis

### 4.1. β-[^11^C]-L-DOPA, 6-[^18^F]Fluoro-L-DOPA, and [^18^F]Fluorotyrosines

Dopamine (3,4-dihydroxyphenethylamine) is simply not passively permeable through the blood–brain barrier (BBB), and its presence and use in the brain requires biosynthesis of the molecule within brain neurons; conceptually, then, using a radiotracer that imaged a step in the biosynthesis was recognized as a possibly useful approximation of dopamine concentrations. Ideally, that would be a tracer of tyrosine hydroxylase (TH), the enzyme that converts tyrosine to L-3,4-dihydroxyphenylalanine (L-DOPA) and is the rate-limiting step in dopamine biosynthesis: unfortunately, development of a radiotracer to measure TH activity has never succeeded. The alternative approach was to target amino acid decarboxylase (AADC), the enzyme responsible for conversion of DOPA to dopamine: although AADC is common to all monoaminergic neurons, in certain brain regions such as the striatum it is predominantly found in dopaminergic neurons. The development of such radiotracers began with 5-[^18^F]fluorodopa (5-[^18^F]fluoro-L-3,4-dihydroxyphenylalanine) prepared in the early 1970s by the radiochemistry group at McMaster University in Canada, but translation to human imaging was realized in 1983 [4] with the positional isomer 6-[^18^F]fluorodopa (6-[^18^F]FDOPA, 6-[^18^F]fluoro-L-3,4-dihydroxyphenylalanine) (Figure 2). 6-[^18^F]FDOPA remains routinely used for human studies today.

The in vivo fate of 6-[^18^F]FDOPA is complex. Transport from the bloodstream into the CNS is via one or more amino acid transporters (e.g., L-type amino acid transporter, LAT1) [5], where the intracellular enzyme AADC acts to form 6-[^18^F]fluorodopamine. That newly synthesized molecule is then moved into the storage vesicles by a second specific transporter, the vesicular monoamine transporter type 2 (VMAT2), and over short scan times (<90 min) [^18^F]FDOPA behaves as an irreversibly trapped radiotracer. Most applications using in vivo imaging of 6-[^18^F]FDOPA in the brain utilize a simple analytic scheme of calculating what is termed the incorporation rate (or uptake rate constant) relative to plasma (K_i_) or occipital cortex (K_occ_). The use of extended imaging times (4 h) has been reported as a means to estimate the exocytotic release, metabolism, and clearance of [^18^F]fluorodopamine, providing an indicator of the effective turnover of dopamine (K_loss_) [6].

The use of [^18^F]FDOPA can be complicated by the potential formation of peripheral radiolabeled metabolites through the action of catechol O-methyltransferase (COMT), as the metabolites are BBB permeable and can decrease signal to background ratios, particularly in extrastriatal regions of low uptake. Radiotracers that are not catechols avoid that problem, and 6-[^18^F]fluoro-L-meta-tyrosine ([^18^F]6-FMT, Figure 2) was prepared as an alternative to radiolabeled DOPA: it is not a substrate for peripheral metabolism by COMT, undergoes transporter-facilitated movement across the BBB and is decarboxylated by AADC to form 6-[^18^F]fluorotyramine. That amine is poorly transported by VMAT2 for storage in vesicles [7], but is subject to the actions of intraneuronal monoamine oxidases (MAO): the resulting oxidation product 6-[^18^F]fluoro-3-hydroxyphenylacetic acid (FHPAA) is trapped in the brain tissues. Comparative studies of [^18^F]FDOPA and [^18^F]FMT in animals have supported similar or perhaps increased sensitivity of the latter to losses of dopaminergic terminals [8,9], and limited studies of both radiotracers in the same human subjects also supported better sensitivity which was assigned as better specificity as a measure of AADC [10,11]. However, [^18^F]FMT is not useful for determining K_loss_, the useful measure of dopamine turnover, as the trapped species ([^18^F]FHPAA) is not released via vesicular exocytosis.

The predominate radiotracer for imaging of dopamine synthesis has remained [^18^F]FDOPA, but syntheses of the carbon-11 labeled radiotracers β-[^11^C]-L-DOPA and 6-[^11^C]methyl-*m*-tyrosine were also accomplished. The β-[^11^C]-L-DOPA forms [^11^C]dopamine after AADC-mediated decarbozylation, and provides human brain images and uptake data much like 6-[^18^F]fluorodopa. A study comparing β-[^11^C]-L-DOPA, 6-[^18^F]fluoroDOPA and 6-[^11^C]methyl-*m*-tyrosine in the monkey brain showed highest incorporation (K_i_ values) for the 6-[^11^C]methyl-*m*-tyrosine, but good sensitivity of all three radiotracers to 1-methyl-4-phenyl-1,2,3,6-tetrahydropyridine (MPTP)-induced dopaminergic terminal losses in the striatum [12]. The short half-life of carbon-11 limits the length of the imaging study and effectively prevents distribution of radiopharmaceuticals outside of facilities with on-site medical cyclotrons. Carbon-11 labeled radiotracers might be useful for multi-radiotracer protocols but the 6-[^11^C]methyl-m-tyrosine has not been translated to human imaging, and none of these ^11^C-radiotracers are expected to diminish the potential use or impact of [^18^F]FDOPA imaging in human studies of neurodegenerative diseases.

### 4.2. Future of AADC Radiotracers

[^18^F]DOPA remains the most-used radiotracer for human studies of dopamine biosynthesis, often in clinical studies combining it with measures of other aspects of dopaminergic nerve terminals (the DAT, VMAT2, dopamine receptors or perhaps drug-stimulated dopamine release) [13], and is now used in some clinical settings for the differential diagnosis of parkinsonism [14]. There have been no recent advances in the development of new imaging radiotracers that target steps in the biosynthesis of dopamine. However, the use of [^18^F]FDOPA (and related [^18^F]fluorotyrosines) as radiotracers for oncologic studies has been growing [15]. Finally, recent advances in fluorine-18 radiochemistry have resulted in improved and higher yielding methods for syntheses of these radiotracers for human studies [16,17,18].

## 5. Dopamine Receptors

The chemical syntheses and preclinical studies targeting in vivo dopamine receptor imaging had been pursued for years before the PET imaging of [^11^C]N-methylspiperone in humans in the early 1980s [19]. In the following nearly four decades, there have been reported a large group of radioligands designed not just to image the distribution of receptor binding sites, but to also address receptor subtype selectivity, sensitivity to endogenous dopamine, or utility for drug occupancy studies. In part, this was aided by availability of multiple chemical scaffolds that have been explored during the development of therapeutic drugs for use in neurology and psychiatry.

The introduction of receptor-binding radiotracers into human imaging stimulated the development of data analysis methods that differed significantly from the simple estimates of such as K_i_ for [^18^F]FDOPA and [^18^F]FMT. Using the dynamic time-radioactivity curves obtained from images, together with measures of radiotracer delivery to the brain (obtained from blood sampling), pharmacokinetic modeling was applied to produce estimates of parameters that are proportional to concentrations the target protein (originally receptors, but also applicable to transporters, enzymes, and ion channels) [20]. As radioligand development efforts have matured, the field has moved to using mostly reversibly-binding radiotracers and a variety of simplified pharmacokinetic models, with the outcome measure usually representing a combination of the receptor number and affinity, such as the binding potential (BP = B_max_/K_d_) or distribution volume ratio (DVR = BP + 1). The BP_ND_ is today the most commonly reported value, as it can be calculated with simplified methods using a reference region of the brain for estimating the nondisplaceable (ND) concentration of radiotracer. In a minority of studies, more complicated multi-injection protocols are employed that allow independent estimates of B_max_ and K_d_. For additional information of PET pharmacokinetic modeling the reader is directed to recent reviews [21,22].

### 5.1. Dopamine D2/D3 Receptor Radioligands

In the early days of dopamine receptor radioligand development, most target compounds were described as dopamine D2 ligands. However, with their poor in vitro subtype selectivity combined with the understanding that the D2 and D3 subtypes of dopamine receptors have significant overlap in several regions of the brain, it became more accepted to generalize most of them as D2/D3 radioligands (although some have been termed “D2- or D3-preferring” radioligands). The earliest radiotracers were radiolabeled forms of high affinity antagonists such as the butyrophenone neuroleptics and substituted benzamides, followed in later years by radiolabeled agonists such as [^11^C]N-propylnorapomorphine and [^11^C]PHNO ([^11^C]]-(+)-4-propyl-9-hydroxynaphthoxazine).

The butyrophenone neuroleptics were developed as high-affinity dopamine antagonists for psychiatric use, with haloperidol the molecule selected for commercialization and which has been in regular use for decades in psychiatric medicine. Early synthetic efforts in radiochemistry had selected haloperidol for radiolabeling with fluorine-18, but it proved unsuitable as an in vivo radioligand [23]. Success was found with spiroperidol (spiperone), which in tritiated form had been accepted as a well-characterized in vitro ligand for dopamine receptors [24]: derivatization by simple N-[^11^C]methylation produced the radiotracer [^11^C]N-methylspiperone, which was utilized in 1983 for the PET imaging of dopamine receptors in the human brain [19], a study often credited with being the first human receptor PET imaging study. In subsequent years a modest number of ^11^C-, ^18^F-, and ^76^Br- (T_1/2_ = 16 h) derivatives of spiperone and benperidol were prepared [25], and several advanced to human studies. With the introduction of radioligands based on the substituted benzamide structure that had improved in vivo pharmacokinetic properties, interest in the butyrophenone neuroleptics as in vivo PET radiotracers has significantly decreased, with only a few recent human studies reported [26].

The development of benzamide neuroleptics as in vivo PET ligands starts with sulpiride, the modifications of which led first to remoxipride and then to raclopride and epidepride [27]. Raclopride (Figure 2) is a selective but not specific D2 ligand, and in carbon-11 labeled form it became a very useful in vivo PET radioligand for PET studies in humans. Further structural modifications of epidepride led to the discovery of a [^18^F]fluoropropyl derivative, [^18^F]fallypride (Figure 2), with very high D2/D3 affinities. Both [^11^C]raclopride and [^18^F]fallypride have found widespread use in human studies, but their applications are quite distinct. [^11^C]Raclopride, with a moderate binding affinity, is useful for studying the high concentrations of receptors in the human striatum, and furthermore the good reversibility of [^11^C]raclopride binding has made it useful for studies examining the effects of alterations of endogenous dopamine levels. [^18^F]Fallypride, [^11^C]FLB 457 and related derivatives of eticlopride and epidepride [25], all of which have about 10-fold higher in vitro affinities for D2/D3 receptors than raclopride, have proven more applicable to studies of the 10-fold lower numbers of dopamine receptors in extrastriatal regions. This may be of value for evaluating changes in dopamine D3 receptors in the thalamus.

An alternative approach to imaging of dopamine receptors has been through the radiolabeling of agonist molecules, which have been proposed to offer the potential advantages of increased sensitivity to changes in endogenous dopamine and possibly differential in vivo binding to high vs. low affinity receptor binding sites. The most studied radiotracers have been substituted apomorphines (N-[^11^C]propylapomorphine (NPA) and related) and the compound PHNO (Figure 2). As with the butyrophenones and the benzamide radioligands, none of these agonists have proven to be either a D2 or D3 specific radioligand; some investigators have described PHNO as D3-preferring based on the high binding in the D3-rich globus pallidus [28] and moderately higher D3 affinity for PHNO [29]. The agonist radiotracers do, however, show better sensitivity to amphetamine-stimulated release of dopamine [30,31], resulting in larger decreases of in vivo radiotracer binding than are observed with typical benzamide radioligands like [^11^C]raclopride.

### 5.2. D1 Receptor Radioligands

Despite higher abundance than D2 receptors in some brain regions, including the striatum, the D1 receptor has received far less attention from radiopharmaceutical chemists. A handful of carbon-11 labeled ligands have been prepared based on the benzazepine structure, two of which ([^11^C]SCH 23390 and [^11^C]NNC 112, Figure 2) have seen applications in human studies [32] despite their confounding high affinity for the serotonin 5-HT_2A_ receptors, as the latter are found in low numbers in the striatum. The syntheses of fluorine-18 radioligands for D1 receptors has been even more limited and none of them have not reached human use [25]. D1 radioligands, if and when successfully translated into human studies, should have applications in PET studies of psychiatric diseases [32].

### 5.3. D4 and D5 Receptor Radioligands

As with the D1 receptors, efforts towards the synthesis of radiolabeled D4 antagonists have been limited. Radioligands with high affinity and good selectivity towards the D4 receptor have been synthesized, with some showing encouraging results of good in vitro selectivity for D4 receptors [33] and promising in vivo biodistributions in animal brains [34]. None of these have yet been advanced into human studies.

The dopamine D5 receptor is expressed in low levels in several cortical and extrastriatal regions of the brain. There are currently no D5-selective or specific radioligands reported.

### 5.4. Imaging of Dopamine Receptors: Current and Future Applications

The field of in vivo imaging of dopamine receptors remains exciting and very productive: as a research tool, PET imaging of the dopamine receptors is widely employed for clinical studies in neurology [35,36,37], psychiatry [38], or drug abuse and addiction [39,40]. Receptor imaging radiotracers are most often used to understand the underlying differences in receptor densities in different pathologies, to evaluate the functional differences in dopaminergic neurotransmission in disease, or to measure the occupancy of newly developed therapeutic drugs.

The methods for studies of receptor densities are straightforward, in that they compare in vivo estimates of regional brain concentrations of receptors (e.g., BP_ND_) between patients and appropriately matched normal controls. Such studies have provided intriguing insights into the potential role of dopamine receptors in diseases, but often the results are decidedly mixed. Rarely are studies replicated using the same radioligands, or even with radioligands that have undergone the equivalent preclinical validation tests. Published studies often employ different protocols and patient groups, and there is little if any standardization of methods for performance and analysis of human PET studies. At least a consensus has been reached on the nomenclature and definition of specific binding measures for reversible in vivo radioligands [41], but better validation of radiotracers and standardization of PET techniques might be advantageous. In addition, there is underlying variability in the numbers of receptors in normal brains, and considerable experimental variance in the in vivo PET measures of radioligand binding (BP_ND_, K_d_ or B_max_) [42,43]. Nevertheless, research investigations into the differences in D1, D2, and D3 dopamine receptors in a wide variety of pathologies have been pursued and continue today.

An alternate application of receptor imaging does not emphasize the baseline differences (if any) in numbers of receptors in the brain, but instead employs receptor occupancy studies [44] to study disease-related differences in radioligand binding in response to test conditions expected to raise or decrease dopamine release into the synapse. For such studies, values of specific ligand binding such as BP_ND_ are measured at rest and after a stimulation, most often a pharmacological agent that increases dopamine release (e.g., amphetamine [44] or methylphenidate [45], or an agent that decreases dopamine biosynthesis (α-methyl-p-tyrosine (AMPT), an inhibitor of TH). A decrease in BP_ND_ is then taken as an index of receptor occupancy due to stimulated dopamine [22], and conversely, an increase in BP_ND_ is assigned as higher availability of receptors unoccupied by dopamine. Such studies of dopamine occupancy of receptors have been extended to addiction (drugs, gambling, eating, smoking) and for studies such as mental stimulation (video games) [46]. Sensitivity to changes in dopamine release or biosynthesis is not a uniform trait of dopamine receptors, nor the same for all radioligands: whereas the D2 ligands [^11^C]raclopride, [^11^C]NPA and [^11^C]PHNO have been useful, other radioligands such as the butyrophenone neuroleptics (e.g., radiolabeled spiperones) and D1 receptor ligands ([^11^C]SCH 23390) show either no or paradoxical responses to manipulations of dopamine [44,47,48].

Finally, in vivo dopamine receptor studies have found a place in therapeutic drug development, including the evaluation of new antipsychotic drugs. PET imaging of D2 receptors was an important tool in determining the occupancy of receptors (approx. 70%) needed for therapeutic effect, but without causing extrapyramidal symptoms [49]. Since that original study, testing of new candidate drugs for in vivo D2 receptor occupancy has been accepted as part of overall evaluation of suitability for clinical use [50].

The use of dopamine receptor radioligands for in vivo imaging is thus now quite common, but there remains a need for radioligands that are truly specific for their intended binding sites [51], particularly for the D2 and D3 subtypes, such that accurate estimates of changes in each subtype might be made in the brain regions that have considerable overlap in concentrations. The definition of “specificity” might be questioned: certainly, a 1000-fold difference of in vitro binding affinities might denote specificity, but such large differences have been difficult to obtain for D2 or D3 radioligands. Some of the best, such as [^11^C/^18^F]N-methylbenperidol (200-fold D2 selectivity) and [^18^F]fluortriopride (159-fold D3 selective) are still described as selective radioligands. Current efforts at the synthesis of radioligands with improved D2 or D3 selectivity are directed at substituted phenylpiperazines related to the atypical antipsychotic aripiprazole [51,52,53]. 

There are also questions remaining as to the utility of in vivo radioligands to answer fundamental questions of dopaminergic receptor biochemistry. Dopamine receptors are known to undergo agonist-induced internalization, and that has been proposed as the explanation for the persistent decrease of in vivo binding of [^11^C]raclopride after amphetamine administration, even after dopamine levels have returned to normal. The opposite and paradoxical effect on in vivo binding of [^11^C]spiperone [48] has suggested that some antagonist radioligands can bind to both surface and internal binding sites, but radiotracers that are agonists do not [44]. It remains unclear if internalization is important, or the same, in different pathological conditions [54]. There are also the issues of the affinity states of G-protein linked receptors: D1, D2, and D3 receptors are proposed to exist in vivo in both high- and low-affinity states. The potential for in vivo radioligands to discriminate between the two affinity states remains unsettled [55,56,57]. Most human PET studies with receptor ligands do not attempt the complicated experiments necessary to separately estimate both affinity (K_D_) and concentration (B_max_). Terms that include both (e.g., binding potential: BP = B_max_/K_D_) [20,41] are widely employed and investigators assume binding affinities remain constant. In reality, any observed changes of in vivo radioligand binding BP values can result from a change in either parameter. Better analytical methods—or radioligands more clearly sensitive to the status of receptor linkage to G-proteins—will be needed to further investigate the affinity states of dopamine receptors.

## 6. Neuronal Membrane Dopamine Transporters (DAT)

The neuronal membrane dopamine transporter (DAT) functions to recover dopamine from the synaptic cleft, and once delivered into the cytosol it is made available for repackaging into vesicles, or subject to catabolism. It was thus proposed as a possible in vivo marker of presynaptic dopaminergic terminal density. After early investigations of radiolabeled forms of drugs such as nomifensine (2-methyl-4-phenyl-1,2,3,4-tetrahydroisoquinolin-8-amine) and GBR 12909 (1-(2-[bis(4-fluorophenyl)methoxy]ethyl)-4-(3-phenylpropyl)piperazine), most development and application efforts have centered on two options: radiolabeled phenyltropanes ([^11^C]CFT and related ^11^C-and ^18^F-labeled synthetic 3-phenyltropane analogs) and [^11^C]methylphenidate.

Cocaine is an inhibitor of the DAT and was radiolabeled by both O- and N-[^11^C]methylation, and although demonstrated in humans to bind to the DAT site in vivo [58], its rapid pharmacokinetics and metabolic N-dealkylation limited its application as a routine PET imaging agent. Better pharmacokinetics, and a wide variety of opportunities for radiolabeling, were found for synthetic compounds based on the 3-phenyltropane structure, with first the carbon-11 labeling of WIN 35,428 (also known as β-CFT), a compound already in use as a tritiated in vitro radioligand for the DAT. Human PET studies of [^11^C]CFT demonstrated high accumulations in regions of dopaminergic innervation, and good pharmacokinetics suitable for modeling to obtain estimates of transporter numbers in normal and disease conditions. Over the decades since introduction of [^11^C]CFT in the early 1990s, a very large number of analogs have been synthesized which incorporate ^11^C or ^18^F [59]. Several of the fluorine-18 labeled 3-phenyltropanes (e.g., [^18^F]FE-PE2I; Figure 2) [60,61] have been translated into human studies, and together with continued use of [^11^C]CFT, provide DAT radioligands for numerous clinical research studies.

Methylphenidate (Ritalin^®^, Figure 2) is a drug routinely used in psychiatric medicine. The radiolabeling of methylphenidate was relatively simple, requiring only ^11^C -methylation of an appropriate nor-methyl precursor [62], and the clinical history of the drug allowed for its rapid translation into human imaging studies. No successful ^18^F-labeled methylphenidate derivative has been developed, but [^11^C]methylphenidate is very useful for multitracer PET studies.

### Imaging of the DAT: Current and Future Applications

As with the dopamine receptors, the imaging of the DAT has found use for research studies in neurology (Parkinson’s and other movement disorders), psychiatry (schizophrenia, ADHD), addiction, drug abuse (cocaine, methamphetamine), and new drug development. These studies are all done using determinations of binding site availability (usually BP values), and in contrast to the dopamine receptors, the DAT is not a site utilized for studies of DA concentrations or release. As there are no uniform methods for evaluation of new radiopharmaceuticals, and seldom any direct comparison of two or more radiotracers, no single PET radiotracer for the DAT has progressed to true widespread use. SPECT (single-photon emission computed tomography) imaging of the DAT is currently approved for clinical use in many countries, employing the radiotracer DaTScan^®^ (Ioflupane I123 Injection: N-ω-fluoropropyl-2β-carbomethoxy-3β-(4-[^123^I]iodophenyl) nortropane), but the advantages inherent in PET imaging (resolution, quantification) should encourage the eventual advancement of one or more of the fluorine-18 labeled DAT radioligands into regulatory approval and commercial availability [61].

## 7. Vesicular Monoamine Transporters (VMAT2)

The vesicular monoamine transporter type 2 (VMAT2) is a vesicle membrane-bound presynaptic protein responsible for the movement of dopamine (and other monoamine neurotransmitters) from the cytosol into the storage vesicles. Chronologically, the VMAT2 site was the last part of the dopamine system to be successfully imaged in humans. This may in part have been to the paucity of candidate molecules available for radiolabeling, but starting with the preparation of [^11^C]tetrabenazine and successful imaging in humans in 1993, a series of carbon-11 and fluorine-18 labeled benzisoquinolines were prepared and evaluated in human imaging studies [63]. Human PET studies have predominantly utilized [^11^C]dihydrotetrabenazine ([^11^C]DTBZ, Figure 2) a purposefully designed high affinity metabolite of tetrabenazine, with increasing use in recent years of the molecule [^18^F]fluoropropyl-dihydrotetrabenazine ([^18^F]AV-133) (Figure 2). Additional radiolabeled tetrabenazine derivatives were synthesized but none placed into human studies [64]. [^11^C]DTBZ and [^18^F]AV-133 are examples of compounds that are specific for a single binding site (VMAT2) but not for specific nerve terminals, as the VMAT2 is common to all monoaminergic neurons. However, it is of practical use for imaging the very high and predominant concentration of dopaminergic nerve terminals in the human striatum. The VMAT2 binding site was originally proposed as a site that was less susceptible to regulation, and that radioligands for the VMAT2 would be less sensitive to changes in endogenous dopamine. Those expectations were not completely met, as the in vivo binding of VMAT2 radioligands was shown to be increased upon severe depletion of dopamine levels due to pharmacological intervention (AMPT) [65] or disease (Dopa-responsive dystonia [66]). At present, no further investigations into new VMAT2 radioligands is evident, and there is only limited research into new chemical structures that might form the basis for new radiotracer development [67].

## 8. Monoamine Oxidases (MAO): Inhibitors and Substrates

MAO A and B are intracellular flavoenzymes bound to the outer membrane of mitochondria found throughout the neurons and astrocytes of the human brain. Although MAO inhibitors are valuable therapeutic agents for disorders such as Parkinson’s disease, they are not candidates for specific dopaminergic imaging agents due to this widespread distribution. Nevertheless, they are an important component of the dopamine system, and have been the target for in vivo PET radiotracer development [68,69]. There have been three approaches to MAO radioligands for PET: suicide inhibitors, reversible antagonists, and metabolic substrates.

The suicide inhibitors deprenyl and chlorgylline were synthesized in carbon-11 form and demonstrated to be trapped in human brain in proportion to MAO levels, with [^11^C]deprenyl showing MAO-B selectivity and [^11^C]chlorgylline MAO-A selectivity [70]. Improved pharmacokinetics and sensitivity were achieved by selective deuteration of [^11^C]deprenyl, and ^18^F-labeled analogs (e.g., [^18^F]fluorodeprenyl and [^18^F]rasagaline), were synthesized in subsequent years [68]. [^11^C]Deprenyl and its deuterated form have remained the most utilized MAO radiotracers.

The extensive medicinal chemistry efforts in MAO inhibitors intended for therapeutic use provided many additional options for new PET radiotracer development, resulting in the preparation of carbon-11 and fluorine-18 labeled radiotracers that function as reversible inhibitors (rather than the suicide inhibitors like deprenyl). Several of these have been successfully introduced into human studies, among them [^11^C]harmine and [^11^C]befloxatone [68].

Finally, a few efforts have been made to apply the concept of metabolic trapping to the design of MAO imaging agents. For this approach, radiotracers were selected or designed to be BBB permeable and subject to metabolism by MAO in brain tissues, with the resultant radiolabeled metabolites irreversibly trapped in the tissues in proportion to the enzymatic activity. None of these potential MAO radiotracers have yet been taken to human studies, but they might provide a different measure of MAO enzymatic activities than the radiolabeled enzyme inhibitors [69].

## 9. Putting It All Together: Multitracer Studies?

There are now quite good in vivo PET radiotracers for most of the biochemical steps of dopaminergic neurotransmission except for TH, and the obvious question is: why not use PET to study all of the steps in the same subjects? There are many examples of dual radiotracer studies but the performance of multitracer studies using three or four different radiotracers is challenging: not many research groups have the ability or inclination to routinely synthesize multiple radiotracers for human studies, particularly as most multitracer studies utilize two or three carbon-11 labeled radiotracers. Protocols for multiradiotracer studies can be complex, and consideration must be made for total radiation exposures of subjects, although use of carbon-11 provides for shorter imaging times and possibly lower radiation doses. Multitracer studies of dopamine systems have been reported by the research group at TRIUMF/University of British Columbia, using the combination of [^18^F]FDOPA with [^11^C]methylphenidate (DAT) and [^11^C]DTBZ (VMAT2). In studies of subjects with Parkinson’s disease, *LRRK2* (leucine-rich repeat kinase 2) mutations, or Dopa-responsive dystonia, they produced unique insights into the potential differential losses of DA synthesis and storage, DAT binding, and VMAT2 binding related to disease progression [71], age of onset and severity of disease [72], compensatory processes [73,74], and sensitivity to dopamine depletion [66]. In a single patient with Dopa-responsive dystonia, that group performed a four radiotracer study ([^18^F]FDOPA, [^11^C]methylphenidate, [^11^C]DTBZ, [^11^C]raclopride), but no larger studies using four dopaminergic radiotracers have been published. There are, however, numerous published studies of three or four radiotracers completed in single subjects, combining one or more of the dopaminergic radioligands in Table 1 with PET imaging agents for other neurotransmitter systems (serotonin, acetylcholine), amyloid imaging, and glucose metabolism ([^18^F]FDG) [64].

## 10. Perspectives

The field of in vivo PET imaging of the dopaminergic system of the human brain is approaching the age of 40 years. In that time, a wide selection of radiotracers has been invented that target nearly all of the biochemistry associated with the synthesis, action, and degradation of dopamine. From the early demonstrations of specific localization of radioactivity following intravenous administration of the radiotracers [^18^F]FDOPA and [^11^C]N-methylspiperone, we have progressed to the current acceptance of in vivo imaging as a valuable research tool to study the biochemistry of the living human brain in health and disease. This has required achievements in a number of areas—improvements in imaging instrumentation, software for image analysis, and better analytical methods and pharmacokinetic models to extract useful biochemical parameters—but it has been the development of the new medicines (radiopharmaceuticals) that has provided the opportunities for advancement. What does the future hold for the use of PET imaging in patient care? Although several PET radiopharmaceuticals have now received approval by governmental drug regulatory agencies worldwide as safe and effective drugs and have entered the medical marketplace, their development for PET imaging in humans did not follow the traditional pathways for new drugs. The majority of such radioactive drugs were and continue to be initially developed in academic institutions, although in recent years radiochemistry groups have been organized within several major pharmaceutical companies. On one hand, this academic-driven approach has perhaps greatly increased the numbers of researchers and laboratories engaged in this particular type of drug research. On the other hand, it may have inhibited commercial development, as the questions of intellectual property rights surrounding new chemical entities have persisted.

The future clinical impact of PET imaging of the brain dopamine system is thus difficult to predict. The use of PET imaging as a stand-alone diagnostic method has been limited, but the results from imaging studies might in a clinical setting be combined with other measures (e.g., clinical exams, neuropsychological testing, blood chemistry, functional MRI) to better inform the clinician and lead to more appropriate diagnosis and treatment [14]. As a radiopharmaceutical chemist who has been part of the last 40 years of development of PET into a valuable tool in both research and patient care, the author of this review continues to be excited by the opportunities that abound in the design, synthesis, and application of new radiotracers to study the in vivo neurochemistry of dopamine and other neurotransmitters in the human brain. PET imaging has opened wide the window into the human brain, and it is up to us to peer through it.

## Figures and Tables

**Figure 1 biomedicines-09-00108-f001:**
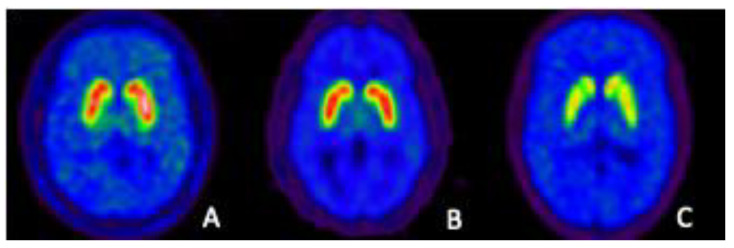
Representative images of specific binding (pixel-by-pixel images of DVR = BP + 1) for three dopaminergic PET radioligands in the striatum of the normal human brain. Note the similarity of the images that are for very different biochemical targets: (**A**) dopamine D2/3 receptors using [^11^C]raclopride; (**B**) vesicular monoamine transporters 2 (VMAT2) using [^11^C]dihydrotetrabenazine ([^11^C]DTBZ); (**C**) the neuronal membrane dopamine transporter DAT using [^11^C]methylphenidate.

**Figure 2 biomedicines-09-00108-f002:**
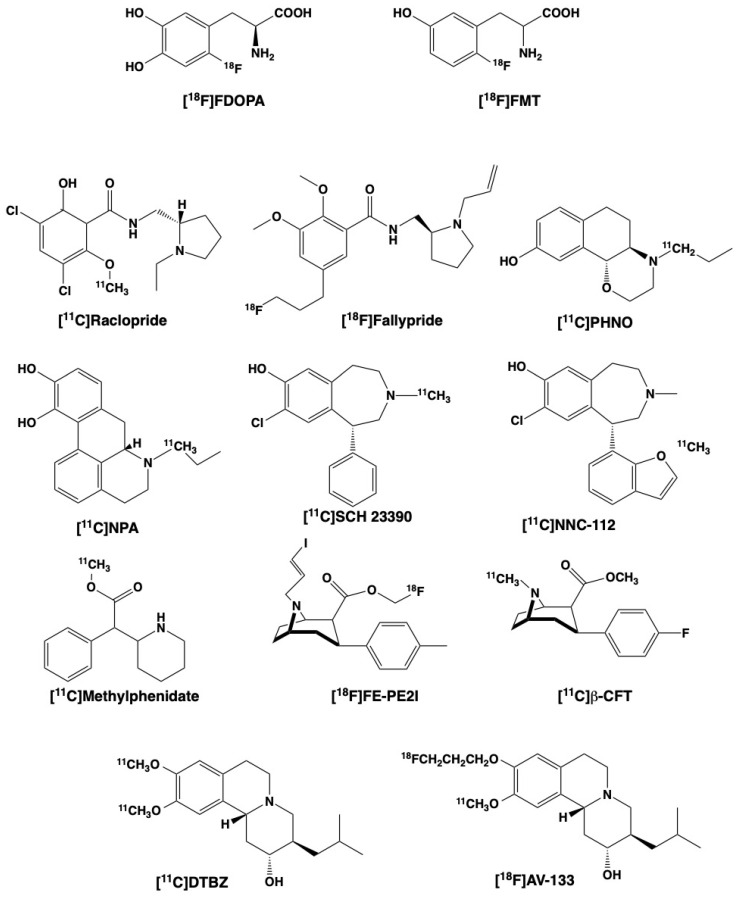
Chemical structures and radiolabeling positions for commonly used dopaminergic PET radiotracers.

**Table 1 biomedicines-09-00108-t001:** Representative radiotracers used for in vivo PET imaging of the neurochemistry of dopamine in the human brain.

Tyrosine Hydroxylase—No In Vivo Radiotracers Available	
**Aromatic Amino Acid Decarboxylase (AADC)**	
β-[^11^C]DOPA	6-[^18^F]Fluoro-m-tyrosine (FMT)
6-[^18^F]Fluorodopa (FDOPA)	6-[^11^C]methyl-m-tyrosine
**D1 Dopamine Receptors**	
[^11^C]SCH 23390	[^11^C]NNC-112
**D2/D3 Dopamine receptors**	
N-[^11^C]Methylspiperone	[^11^C]n-Propylapomorphine (NPA)
[^11^C]Raclopride	[^11^C]PHNO
[^18^F]Fallypride	[^11^C]FLB 457
**Neuronal Membrane Dopamine Transporter (DAT)**	
[^11^C]CFT, [^18^F]CFT	[^18^F]PE2I
[^11^C]Methylphenidate	[^18^F]FE-PE2I
**Vesicular Monoamine Transporter type 2 (VMAT2)**	
[^11^C]Dihydrotetrabenazine (DTBZ)	[^18^F]FluoropropylDTBZ (AV-133)
**Monoamine Oxidases (MAO)**	
[^11^C]deprenyl	[^11^C]harmine
d_6_-[^11^C]deprenyl	[^11^C]befloxatone
[^11^C]chlorgylline	

Abbreviations: DOPA, L-3,4-dihydroxyphenylalanine; SCH 23990, R(+)-7-chloro-8-hydroxy-3-methyl-1-phenyl-2,3,4,5-tetrahydro-1H-3-benzazepine hydrochloride; NNC-112, (+)-8-chloro-5-(7-benzofuranyl)-7-hydroxy-3-methyl-2,3,4,5-tetrahydro-1H-3-benzazepine; PHNO, (+)-4-propyl-9-hydroxynaphthoxazine; CFT, (-)-2β-Carbomethoxy-3β-(4-fluorophenyl)tropane; FLB 457, (S)-5-bromo-N-[(1-ethyl-2-pyrrolidinyl)methyl]-2,3-dimethoxybenzamide; PE2I, (E)-N-(3-iodoprop-2-enyl)-2-beta-carbomethoxy-3-beta-(4′-methyl-phenyl)nortropane; FE-PE2I, (E)-N-(3-iodoprop-2-enyl)-2-beta-carbofluoroethoxy-3-beta-(4′-methyl-phenyl)nortropane.

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
