# Peer review of "11C- and 18F-Radiotracers for In Vivo Imaging of the Dopamine System: Past, Present and Future"

_biomedicines, 2021, doi:10.3390/biomedicines9020108_

Round 1

Reviewer 1 Report

Comments on Review: Radiotracers for in vivo Imaging of the Dopamine ...

The manuscript by M. Kilbourn is a well written and structured review, skillfully condensing the voluminous amount of existing publications on the addressed topic with enough distance for clarity. It reflects precisely and comprehensively the present status of in vivo imaging of the dopaminergic system, and along the way, how to develop radiotracers, at which the biochemical and pharma-cological basis must not be neglected. It is thus very instructive, sometimes even educative, for both experienced and new readers interested in that topic, giving as well an outlook on necessities and possibilities in the future.

The article as it is now, could be published besides some minor changes for clarity and a few corrections to be made (see below). Following aspects and suggestions, however, may still be considered, which may further improve the manuscript: title, figures and tables, references.

The title is very general and appears as that of a book. It may be modified and in particular be specified, reflecting the actual contents of the review. There is a huge amount of publications, addressing system radiotracers and radioligands for the dopamine, which are labeled with other positron- and gamma-emitters, as well as research and clinical studies with SPECT, but the review is only focused on the PET studies with carbon-11 and fluorine-18. The actual content of the article thus may be more accurately described, for example by: 11C- and 18F-Radiotracers for in vivo Imaging of the Dopamine System by PET: Present and Future”.

The review would probably also gain, if the author could provide a few more figures, schemes and/or tables. Some pictures, perhaps also from preclinical studies, could illustrate more targets than the three selected in the given figure 1; e.g., a table could summarize those tracers and ligands together with their indications, which are established in clinical routine.

A great job was done by the author, selecting essential references from a huge wealth of existing ones. Still some more of following suggested, however, might be considered to be adjoined upon checking.

On one hand some general ones might be added. For example, in context with the optimal log P-range for a BBB-permeability: Waterhouse R.N. Mol. Imaging Biol. 2003,5:376-89, or with respect to the past (as indicated in the title) and historic dates, for example on PET-development: Jones T.; Townsend D. J. Med. Imag. 2017, 4:011013. Also an early monograph from 1991 on the same topic as the submitted article might be cited, entitled “Brain Dopaminergic Systems: Imaging with Positron Tomography” by Baron et al.(eds.) in the series: Developments in Nuclear Medicine, Kluwer Academic Publishers, Dordrecht, Boston, London, vol. 20. In the same series (vol. 23) is also a review by H.H. Coenen on the “Biochemistry and evaluation of fluoroamino acids”, in: PET Studies on Amino Acid Metabolism and Protein Synthesis (Mazoyer B.M. et al., eds.) 1993, 109–129. There, the biosynthesis and metabolism of 18F-labeled amino acids and derivatives are addressed as “false” precursors of neurotransmitters or as potential enzyme inhibitors of AADC or MAO with relation to PET-imaging of the dopamine system, documenting already the 1980th as ‘decade of [18F]fluoro-amino acid research’. [If not available, this book chapter can be provided.]

On the other hand, several reports on preclinical findings and results with respect to D3, DAT and MAO-A targets appeared during the last decade, providing important insights, which are probably worth to be mentioned in the review article:

Some are missing, that could provide a better understanding for the in-vivo failure of some “D3-preferring” radioligands to the reader. For instance, Hocke et al. (Nucl Med Biol 2014, 41:223-228) demonstrated that 18F-labeled D3 ligands (derived from BP897) had a globally high total distribution volume in rat brain, impairing the determination of D3-specific binding there. Interestingly, very high displaceable binding of benzamide ligands was observed in the D3 receptor-rich pituitary gland, located outside of the brain.
Similarly, Fehler et al. (Chemistry 2014, 20:370-375) and Nebel et al. (Molecules 2016, 21, 1144) developed a D3-selective 18F-labeled azocarboxamide that is highly suitable to determine the D3 receptor density in vitro by autoradiography. However, its high uptake in the ventricular system (specific D3-binding to the ependymal cell layer) together with specific uptake in the pituitary gland prevented PET imaging studies of rat brain regions.

Concerning DAT ligands that facilitate preclinical PET imaging, also citation of important work is missing. Cumming et al. (J Cereb Blood Flow Metab 2014; 34:1148-1156) developed ¹⁸F-FP-CMT, a phenyltropane with superior properties for quantitative imaging of DAT in rat brain. ¹⁸F-FP-CMT has only a moderate affinity to DAT, but it is highly DAT-selective, allowing quantitative PET studies obtained with recordings lasting only 45 minutes.

Considering tracers for MAO-A, the development of [18F]fluoroethyl-harmol is neglected, that has been successfully studied by compartmental analysis of binding in living rat brain (see: Maschauer et al. J Neurochem 2015;135:908-917, and Cumming et al. Synapse 2015, 69:57-59).

Surprisingly, 3 references, no. 16 – 18, on improvements of 18F-labeling are given, while nearly all others concern properties and behavior of tracers and ligands. Of course, efficient labeling is a key issue, but if those should be kept, no. 16 might be exchanged by an article from the same group: Zischler J. et al. Alcohol-Enhanced Cu-Mediated Radiofluorination. Chem. Eur. J. 2017, 23:3251–3256. The latter describes an improvement of the Cu-mediated radiofluorination method, also addressed in refs. 17 and 18, while the publication of Zlatopolskiy et al. disproves advantages and even practicality of the Pt- or Pd-catalyzed radiofluorination methods.

Comments on necessary corrections and optional minor changes:

Page 1, line 20/21(1;20/21): Better replace “50+” by “almost 60 years”.

1; 34/35: Please extend the sentence: … used by … research laboratories “and for patient care” (or: “and in clinical settings”) around the world. Otherwise the impression is caused that there are just research studies done with PET, while on page 7 it is clearly stated: “… widespread use in humans …”!

1; 40: Most radiotracers “are” small molecules …

2; 58: Better write “should not” than “must not”? It may not happen rather than not be done.

2; table, line 8: The superscript “†” is not explained in a footnote or elsewhere.

2; table, last line: Please check superscript “11C” and terms of all nuclide symbols in square brackets (without spacer!) throughout the article.

2; 68: At the start of the new chapter and for relation to the title, the sentence may be reversed: The dopaminergic system in the brain was a …

3; 81: Please define abbreviations when first mentioned throughout the article. (Explanation of VMAT2 is not given before page 5!)

3; 112: “<50 years” seems unrealistic (see hist. references given above), should rather read “almost 50 years”.

4; chapter 4.1: Give positional numbers of substituents (correct 6-fluoro Dopa) and stereochemical information (correct L-Dopa) at least when compounds are first mentioned in the text. Now first use only on page 6!

4; 132: Check spacers (mostly superfluous) in radionuclide symbols, e.g. [18F], and before new sentences etc. throughout the article.

5; figure 2: Stereochemistry is not indicated for the first two structures. Moreover the first structure is not FDOPA (the standard abbreviation for 6-[18F]fluoro-3,4-dihydroxy-L-phenylalanine) but 3,4-dimethoxy-5-fluoro-D/L-phenylalanine. Please also check the structure or abbreviation of FMET. Given here is 4-methoxy-5-fluoro-D/L-phenylalanine, i.e. meta(5)-fluoro-O-methyl-D/L-tyrosine.

6; 156: Better replace “the oxidation” by “its oxidation”.

6; 169: Better substitute “11C” by “carbon-11”.

6; 180: Better replace first “developments” by “advances”, or similar.

6; 184: Perhaps better replace “much” by “promising”; see comment on ref. 16 above.

7; 221: Define PHNO here, not only on page 8.

8; 251: Sentence not quite clear. … and ‘have/are’ in particular a means …

8; 273: Perhaps change to: … radioligands ‘for D1 receptors’ has been even more limited, and the ‘latter’ have …

8; 278-280: Meaning not quite clear; a mixing up of D1 and D4?

8; 288: Better to use here ‘or drug’ than ‘and drug’?

8; 297: … using the same …

8; 298: The critical remark is very appreciated. Besides the often missing procedures ‘repetition’ and ‘standardization’, also ‘validation’ may be addressed and recommended in this context.

9; 299: New sentence: … PET studies. Those would perhaps be …?

9; 329: Here a pertinent reference (of a review?) is missing, documenting the many … tested for clinical use!

10; 354: Better start new sentence after: … (Bmax). Terms that …

10; 389. Correct the number of the sub-chapter to 6.1 (not 5.4?), and increase all following chapter numbers by one to: 7., 8.,9., and 10.

10; 401: First mentioning of quantification in context with PET! This is perhaps an item for the introduction or even the title. So far, there was just “extraction of numerical data” mentioned.

11; 419: Better make a new sentence after ‘neurons. However, …’

11; 443: A reference would be great after ‘years’.

11; 444: Perhaps a new paragraph may start here with: “The extensive …” to separate the parts on suicide and reversible inhibitors.

12; 459/460: Perhaps better replace brackets by comas and the colloquial “we’re” by “besides” or “although”.

12; 465/466: Text could somewhat be more elaborated with respect to carbon-11, considering health protection problems and the possibility of studies in shorter interval compared to fluorine-18.

12; 483: Isn’t 40 years more correct than 50? See ref. 19 from 1983!

Author Response

I thank the reviewers for a thorough job finding needed corrections in the manuscript, and providing some useful and interesting suggestions.

The point-by-point response is below.  I incorporated every correction, and some of the suggestions.  I apologize about the attempt to identify where changes were made (Page and line numbers of my revision), it was only until after I had finished this that I discovered that each time I closed and re-opened the text document, the Word program renumbered the lines (and I have no idea why, or any control)!   But the page and line numbers are close enough to find the changes in the text.

Response to reviewer #1

The manuscript by M. Kilbourn is a well written and structured review, skillfully condensing the voluminous amount of existing publications on the addressed topic with enough distance for clarity. It reflects precisely and comprehensively the present status of in vivo imaging of the dopaminergic system, and along the way, how to develop radiotracers, at which the biochemical and pharma-cological basis must not be neglected. It is thus very instructive, sometimes even educative, for both experienced and new readers interested in that topic, giving as well an outlook on necessities and possibilities in the future.

The article as it is now, could be published besides some minor changes for clarity and a few corrections to be made (see below). Following aspects and suggestions, however, may still be considered, which may further improve the manuscript: title, figures and tables, references.

The title is very general and appears as that of a book. It may be modified and in particular be specified, reflecting the actual contents of the review. There is a huge amount of publications, addressing system radiotracers and radioligands for the dopamine, which are labeled with other positron- and gamma-emitters, as well as research and clinical studies with SPECT, but the review is only focused on the PET studies with carbon-11 and fluorine-18. The actual content of the article thus may be more accurately described, for example by: 11C- and 18F-Radiotracers for in vivo Imaging of the Dopamine System by PET: Present and Future”. 

Pg 1, Line 2     I agree with the this change in the title, if allowed by the Journal at this stage

The review would probably also gain, if the author could provide a few more figures, schemes and/or tables. Some pictures, perhaps also from preclinical studies, could illustrate more targets than the three selected in the given figure 1; e.g., a table could summarize those tracers and ligands together with their indications, which are established in clinical routine.

As stated in the Introduction, this was never intended as a comprehensive review – there is simply too much material to cover.  Minimal discussion was be extended to preclinical studies, and thus no animal PET scans are included.  For human PET scans, the three chosen were available (without dealing with copyright issues) and were used to demonstrate the point that tracers used to mark dopaminergic terminals provide visually similar images, but represent distinct biochemical interpretations. 

A great job was done by the author, selecting essential references from a huge wealth of existing ones. Still some more of following suggested, however, might be considered to be adjoined upon checking.

On one hand some general ones might be added. For example, in context with the optimal log P-range for a BBB-permeability: Waterhouse R.N. Mol. Imaging Biol. 2003,5:376-89, or with respect to the past (as indicated in the title) and historic dates, for example on PET-development: Jones T.; Townsend D. J. Med. Imag. 2017, 4:011013. Also an early monograph from 1991 on the same topic as the submitted article might be cited, entitled “Brain Dopaminergic Systems: Imaging with Positron Tomography” by Baron et al.(eds.) in the series: Developments in Nuclear Medicine, Kluwer Academic Publishers, Dordrecht, Boston, London, vol. 20. In the same series (vol. 23) is also a review by H.H. Coenen on the “Biochemistry and evaluation of fluoroamino acids”, in: PET Studies on Amino Acid Metabolism and Protein Synthesis (Mazoyer B.M. et al., eds.) 1993, 109–129. There, the biosynthesis and metabolism of 18F-labeled amino acids and derivatives are addressed as “false” precursors of neurotransmitters or as potential enzyme inhibitors of AADC or MAO with relation to PET-imaging of the dopamine system, documenting already the 1980th as ‘decade of [18F]fluoro-amino acid research’. [If not available, this book chapter can be provided.]

The Waterhouse reference can be found in the Pike reference cited (ref #2), did not feel needed to be included here.

I judged there were sufficient general references: general concepts of PET (Hooker and Carson, ref#21), dopamine receptors (ref#25), DAT (Riss, ref #59), VMAT2 (Kilbourn, ref#64), MAO (Kersemans, ref#68),

On the other hand, several reports on preclinical findings and results with respect to D3, DAT and MAO-A targets appeared during the last decade, providing important insights, which are probably worth to be mentioned in the review article:

Some are missing, that could provide a better understanding for the in-vivo failure of some “D3-preferring” radioligands to the reader. For instance, Hocke et al. (Nucl Med Biol 2014, 41:223-228) demonstrated that 18F-labeled D3 ligands (derived from BP897) had a globally high total distribution volume in rat brain, impairing the determination of D3-specific binding there. Interestingly, very high displaceable binding of benzamide ligands was observed in the D3 receptor-rich pituitary gland, located outside of the brain.
Similarly, Fehler et al. (Chemistry 2014, 20:370-375) and Nebel et al. (Molecules 2016, 21, 1144) developed a D3-selective 18F-labeled azocarboxamide that is highly suitable to determine the D3 receptor density in vitro by autoradiography. However, its high uptake in the ventricular system (specific D3-binding to the ependymal cell layer) together with specific uptake in the pituitary gland prevented PET imaging studies of rat brain regions.

Concerning DAT ligands that facilitate preclinical PET imaging, also citation of important work is missing. Cumming et al. (J Cereb Blood Flow Metab 2014; 34:1148-1156) developed ¹⁸F-FP-CMT, a phenyltropane with superior properties for quantitative imaging of DAT in rat brain. ¹⁸F-FP-CMT has only a moderate affinity to DAT, but it is highly DAT-selective, allowing quantitative PET studies obtained with recordings lasting only 45 minutes.

Considering tracers for MAO-A, the development of [18F]fluoroethyl-harmol is neglected, that has been successfully studied by compartmental analysis of binding in living rat brain (see: Maschauer et al. J Neurochem 2015;135:908-917, and Cumming et al. Synapse 2015, 69:57-59).

All of these suggestions are interesting, but I emphasized in this review those radiotracers that had reached human use.  There are far too many “promising” radioligands reported to include. It’s a big, difficult step to go from animals to humans, and many radioligands fail that step, even though they looked so promising in animal models.  A sentence was added at the end of section 3 (around bottom of page 3, maybe top of page 4) emphasizing that research into radiotracers continues, and encouraging the reader to access the literature to learn more.

Surprisingly, 3 references, no. 16 – 18, on improvements of 18F-labeling are given, while nearly all others concern properties and behavior of tracers and ligands. Of course, efficient labeling is a key issue, but if those should be kept, no. 16 might be exchanged by an article from the same group: Zischler J. et al. Alcohol-Enhanced Cu-Mediated Radiofluorination. Chem. Eur. J. 2017, 23:3251–3256. The latter describes an improvement of the Cu-mediated radiofluorination method, also addressed in refs. 17 and 18, while the publication of Zlatopolskiy et al. disproves advantages and even practicality of the Pt- or Pd-catalyzed radiofluorination methods.

The references to the new fluorine-18 radiochemistry were included as they have been a significant advance in radiolabeling; as I am aware of some controversy regarding the credit to be given for the advances in copper-catalyzed fluorinations, I included papers from the different groups. The change for reference 16 was made as suggested (newer and more relevant ref).

Comments on necessary corrections and optional minor changes:

Page 1, line 20/21(1;20/21): Better replace “50+” by “almost 60 years”.  Change made

Pg 1; Line 34/35:  Text changed as suggested Please extend the sentence: … used by … research laboratories “and for patient care” (or: “and in clinical settings”) around the world. Otherwise the impression is caused that there are just research studies done with PET, while on page 7 it is clearly stated: “… widespread use in humans …”!

Pg 1; line 40: Text changed as suggested Most radiotracers “are” small molecules …

Page 2; line 60: Text changed as suggested Better write “should not” than “must not”? It may not happen rather than not be done.

Page 2; table, line 8: Superscript deleted The superscript “†” is not explained in a footnote or elsewhere.

Page 2; table, last line: Corrected Please check superscript “11C” and terms of all nuclide symbols in square brackets (without spacer!) throughout the article.

Page 2; line 70: Text changed as suggested At the start of the new chapter and for relation to the title, the sentence may be reversed: The dopaminergic system in the brain was a …

Text changed as suggested Please define abbreviations when first mentioned throughout the article. (Explanation of VMAT2 is not given before page 5!)

Page 3; 121: Text changed as suggested “<50 years” seems unrealistic (see hist. references given above), should rather read “almost 50 years”.

Page 4; chapter 4.1: Text changed as suggested Give positional numbers of substituents (correct 6-fluoro Dopa) and stereochemical information (correct L-Dopa) at least when compounds are first mentioned in the text. Now first use only on page 6!

Pge 4; 132: Check spacers (mostly superfluous) in radionuclide symbols, e.g. [18F], and before new sentences etc. throughout the article.

All chemical nomenclature double-checked to eliminate the spaces

Page 5; figure 2: Figure replaced, stereochemistry shown for correct structures (embarrassing mistake on my part!) Stereochemistry is not indicated for the first two structures. Moreover the first structure is not FDOPA (the standard abbreviation for 6-[18F]fluoro-3,4-dihydroxy-L-phenylalanine) but 3,4-dimethoxy-5-fluoro-D/L-phenylalanine. Please also check the structure or abbreviation of FMET. Given here is 4-methoxy-5-fluoro-D/L-phenylalanine, i.e. meta(5)-fluoro-O-methyl-D/L-tyrosine.

Page 6; line 179: Text changed as suggested Better replace “the oxidation” by “its oxidation”.

Page 6; line 195: Text changed as suggested Better substitute “11C” by “carbon-11”.

Page 6; line 207: Text changed as suggested Better replace first “developments” by “advances”, or similar.

Page 6; line 211: ‘much’ deleted Perhaps better replace “much” by “promising”; see comment on ref. 16 above.

Pge 7; line 261:name given Define PHNO here, not only on page 8.

Page 8; line 295:sentence reworded Sentence not quite clear. … and ‘have/are’ in particular a means …

Page 8;line 316 : sentence reworded Perhaps change to: … radioligands ‘for D1 receptors’ has been even more limited, and the ‘latter’ have …

Page 8; line 324: corrected to D4 Meaning not quite clear; a mixing up of D1 and D4?

Page 8; line 302: corrected to “or drug” Better to use here ‘or drug’ than ‘and drug’?

Page 8; line 349: corrected … using the same …

Page 8; lines 350-355: This paragraph has been slightly reworded, to include the mentioning of validation The critical remark is very appreciated. Besides the often missing procedures ‘repetition’ and ‘standardization’, also ‘validation’ may be addressed and recommended in this context.

Page 8; lines 350-355: This paragraph has been slightly reworded, to include the mentioning of validation: New sentence: … PET studies. Those would perhaps be …?

9; 329: Here a pertinent reference (of a review?) is missing, documenting the many … tested for clinical use!

Page 9 lines 378-382. A new reference (now #50, Uppoor et al) was inserted in text and ref list.  It better describes how PET imaging can be used in the new drug development process.  Old ref 50 (Farde et al) becomes reference 49.

Pge 10; line 416: Text changed as suggested Better start new sentence after: … (Bmax). Terms that …

Page 10; line 452 and following.Numbering corrected  Correct the number of the sub-chapter to 6.1 (not 5.4?), and increase all following chapter numbers by one to: 7., 8.,9., and 10.

10; 401: First mentioning of quantification in context with PET! This is perhaps an item for the introduction or even the title. So far, there was just “extraction of numerical data” mentioned.

Analysis of data is first mentioned in section 5 (Dopamine Receptors), including refs to modeling.  I have tried not to get into the complex questions of pharmacokinetic modeling, and all the issues surrounding image analysis. Although long involved in such (ref #20), I am far from an expert on the topic!  The reader has been provided with appropriate reviews (refs 21,22)

Page 11; line 485: Text changed as suggested Better make a new sentence after ‘neurons. However, …’

Page 11; line 509: reference inserted A reference would be great after ‘years’.

Page 11; 511: Text changed as suggested Perhaps a new paragraph may start here with: “The extensive …” to separate the parts on suicide and reversible inhibitors.

Page 12; lines 532: Sentence reworded Perhaps better replace brackets by comas and the colloquial “we’re” by “besides” or “although”.

Page 12; lines 539040 text changed as suggested: Text could somewhat be more elaborated with respect to carbon-11, considering health protection problems and the possibility of studies in shorter interval compared to fluorine-18.

Page 12; line 557: Isn’t 40 years more correct than 50? See ref. 19 from 1983!

Agreed, changed

Reviewer 2 Report

This review paper summarizes the development of PET radiotracers targeting various dopamine neurotransmitter systems. At the same time, Dr. Kilbourn presents the perspectives of future development of various dopamine radiotracers. This review paper is well-organized and outlines the development of important dopamine radiotracers. In order to make new radiochemists who may not be familiar with brain dopamine radiotracers understood, the reviewer suggests the following changes.

  1. Figure 2 can be split based on dopamine targets and reassigned by each section. By doing this, Dr. Kilbourn can add more chemical structures of important radiotracers mentioned only in the main text such as β-[11C]DOPA, [11C]FLB457, [11C]CFT, [18F]CFT, and ioflupane (123I).
  2. If appropriate, Dr. Kilbourn might add literature values of IC50 or Ki of each radiotracer particularly in D2 and D3 receptors to give ideas of their selectivity and specificity.

The reviewer also requests to change the minor mistakes in the manuscript.

Line 40: Please add BBB next to blood-brain barrier instead of Line 119

Table 1: DAT, VMAT2, MAO can be added next to each full name.

               [11C]chlorgylline --> [11C]chlorgylline

Line 81: Please provide the full names of DAT and VMAT2 instead of their abbreviation in Line 361 and Line 405, respectively.

Line 117: [11.C]DOPA --> β-[11C]-L-DOPA or β-[11C]DOPA

Line 117: Please mention the evaluation of β-[11C]-L-DOPA. It was mentioned in the title and mentioned in Line 165, but Dr. Kilbourn did not evaluate the tracer. Otherwise, please remove it in the Line 117 and Table 1.

Line 118: The full name of dopamine needs to be mentioned in either Line 71 or Line 78.

Line 156: Spell check --> 6-[18F]fluoro-3-hydrpxyphenylacetic acid

Line 161: Missing the following --> measure of [10,11] (measure of ??? [10, 11])

Line 167-168: Do you mean 6-[11C]methyl-DOPA by 6-[11C]methyl-m-tyrosine?

Line 168: Please provide the full name of MPTP.

Line 197: Please unify the abbreviation either to [18F]FMET or [18F]FMT

              [18F]FMET is used in Table 1 and Line 197

              [18F]FMT is used in Line 151, 157, and 161

Line 228: (24] --> [24]

Line 258: The full name of PHNO need to be moved to Line 221.

Line 263-264: Please check the following sentence --> The agonist radiotracers do, however, show better sensitivity to amphetamine-stimulated dopamine release of dopamine [30,31]

Line 271-272: Please make it clear. serotonin 5HT2A receptors --> serotonin (5HT) 2A receptors or serotonin 2A receptors (5HT2A)

Line 365: The chemical names of nomifensine and GBR12909 might be needed because their structures are not shown.

Line 389: 5.4 --> 6.1

Line 404: Subsequent numbering is wrong. 6 --> 7, 7 --> 8, 8 --> 9, and       9 --> 10

Line 404: As in Line 360, VMAT2 might be added next to the title

Line 429: Monoamine Oxidase: Inhibitors and Substrates --> Monoamine Oxidase (MAO): Inhibitors and Substrates

Line 430: Because MAO is mentioned in Line 156, monoamine oxidase A and B can be changed to MAO A and B.

Line 459: Because TH is mentioned in Line 122, tyrosine hydroxylase --> TH

Line 469: The full name of LRRK2 is needed.

Line 476: There are however numerous --> There are, however, numerous

Author Response

Response to reviewer 2

I thank the reviewers for a thorough job finding needed corrections in the manuscript, and providing some useful and interesting suggestions.

The point-by-point response is below.  I incorporated every correction, and some of the suggestions.  I apologize about the attempt to identify where changes were made (Page and line numbers of my revision), it was only until after I had finished this that I discovered that each time I closed and re-opened the text document, the Word program renumbered the lines (and I have no idea why, or any control)!   But the page and line numbers are close enough to find the changes in the text.

This review paper summarizes the development of PET radiotracers targeting various dopamine neurotransmitter systems. At the same time, Dr. Kilbourn presents the perspectives of future development of various dopamine radiotracers. This review paper is well-organized and outlines the development of important dopamine radiotracers. In order to make new radiochemists who may not be familiar with brain dopamine radiotracers understood, the reviewer suggests the following changes.

  1. Figure 2 can be split based on dopamine targets and reassigned by each section. By doing this, Dr. Kilbourn can add more chemical structures of important radiotracers mentioned only in the main text such as β-[11C]DOPA, [11C]FLB457, [11C]CFT, [18F]CFT, and ioflupane (123I).

I have only added the structure of b-[11C]CFT.  The compounds included in the Fig were chosen to exemplify the variety of chemical structures used, and places for both carbon-11 and fluorine-18 labeling. The reference (Riss, #59) has all of the structures of the tropane series, by far the most used.

  1. If appropriate, Dr. Kilbourn might add literature values of IC50 or Ki of each radiotracer particularly in D2 and D3 receptors to give ideas of their selectivity and specificity.

I gave a lot of thought on providing affinities for compounds.  Unfortunately, the literature is very, very inconsistent – one can find, for any ligand, wide ranges of values.  It is exceedingly hard to find values for different compounds that were obtained in identical fashion (species, tissues, methods).  For the reviewer’s benefit, they can look at such as Cumming et al, Synapse 2011; 65:892-909, which compiled such for dopamine receptors, to see the variety in Kd and Bmax values.  I cannot even find a similar compilation for other targets.  So I felt giving affinity values when there is such variability is only a way to mislead readers – one can selectively choose values to support any conclusion!

The reviewer also requests to change the minor mistakes in the manuscript.

Pg 1 Line 40: Text changed as suggested Please add BBB next to blood-brain barrier instead of Line 119

Table 1: DAT, VMAT2, MAO can be added next to each full name. These were added to table

               [11C]chlorgylline --> [11C]chlorgylline  corrected

Line 81: Please provide the full names of DAT and VMAT2 instead of their abbreviation in Line 361 and Line 405, respectively.

Confused, the text already has full names.  DAT is an acronym used for the neuronal membrane dopamine transporter, and VMAT2 for the vesicular monoamine transporters type 2.   Those are already in the text

Page 4 Line 117:Corrected  [11.C]DOPA --> β-[11C]-L-DOPA or β-[11C]DOPA

Page 6 Line 189-91 Additional text has been added to describe how β-[11C]-L-DOPA is converted to [11C]dopamine, and gives images similar to FDOPA.: Please mention the evaluation of β-[11C]-L-DOPA. It was mentioned in the title and mentioned in Line 165, but Dr. Kilbourn did not evaluate the tracer. Otherwise, please remove it in the Line 117 and Table 1.

Line 118: The full name of dopamine needs to be mentioned in either Line 71 or Line 78.

Disagree.  Dopamine is a term expected to be fully understood by any reader of the Journal. However, full name is given on the first line of section 4.1

Pg 6, Line 179: corrected Spell check --> 6-[18F]fluoro-3-hydrpxyphenylacetic acid

Page 6, Line 184: AADC inserted Missing the following --> measure of [10,11] (measure of ??? [10, 11])

Page Line 193: Corrected to read 6-[11C]methyl-m-tyrosine Do you mean 6-[11C]methyl-DOPA by 6-[11C]methyl-m-tyrosine?

Page 6 Line 194: Name given Please provide the full name of MPTP.

Line 197: Please unify the abbreviation either to [18F]FMET or [18F]FMT

              [18F]FMET is used in Table 1 and Line 197

              [18F]FMT is used in Line 151, 157, and 161

Corrected – [18F]FMT used consistently

Page 7 Line 268: corrected (24] --> [24]

Page 9 Line 374: name moved, and deleted in later section The full name of PHNO need to be moved to Line 221.

Page 8, Line 307: extra “dopamine” deleted Please check the following sentence --> The agonist radiotracers do, however, show better sensitivity to amphetamine-stimulated dopamine release of dopamine [30,31]

Page 8, Line 315: changed to serotonin 5-HT2A Please make it clear. serotonin 5HT2A receptors --> serotonin (5HT) 2A receptors or serotonin 2A receptors (5HT2A)

Page 10, Line 427: Chemical names provided The chemical names of nomifensine and GBR12909 might be needed because their structures are not shown.

Line 389: 5.4 --> 6.1 Numbering corrected

Line 404: Subsequent numbering is wrong. 6 --> 7, 7 --> 8, 8 --> 9, and       9 --> 10 Corrected

Page 11 Line 470: corrected as suggested As in Line 360, VMAT2 might be added next to the title

Page 11 Line 495: corrected as suggested Monoamine Oxidase: Inhibitors and Substrates --> Monoamine Oxidase (MAO): Inhibitors and Substrates

Page 11, Line 496: corrected as suggested Because MAO is mentioned in Line 156, monoamine oxidase A and B can be changed to MAO A and B.

Page 12, Line 532: corrected as suggested Because TH is mentioned in Line 122, tyrosine hydroxylase --> TH

Pge 12 Line 543: full name provided as suggested The full name of LRRK2 is needed.

Page 13 Line 550: corrected as suggested There are however numerous --> There are, however, numerous